# ncRNAs in Therapeutics: Challenges and Limitations in Nucleic Acid-Based Drug Delivery

**DOI:** 10.3390/ijms222111596

**Published:** 2021-10-27

**Authors:** Miguel Hueso, Adrián Mallén, Marc Suñé-Pou, Josep M. Aran, Josep M. Suñé-Negre, Estanislao Navarro

**Affiliations:** 1Department of Nephrology, Hospital Universitari de Bellvitge, 08907 L’Hospitalet de Llobregat, Spain; 2Nephrology and Renal Transplantation Group, Infectious Disease and Transplantation Program, Institut d’Investigació Biomèdica de Bellvitge-IDIBELL, 08907 L’Hospitalet de Llobregat, Spain; amallen@idibell.cat; 3Pharmacy and Pharmaceutical Technology and Physical Chemistry Department, Faculty of Pharmacy and Food Sciences, University of Barcelona, Av. Joan XXIII, 27-31, 08028 Barcelona, Spain; marcsune@ub.edu (M.S.-P.); jmsune@ub.edu (J.M.S.-N.); 4Immunoinflammatory Processes and Gene Therapeutics Lab, Institut d’Investigació Biomèdica de Bellvitge-IDIBELL, 08908 L’Hospitalet de Llobregat, Spain; jaran@idibell.cat; 5Independent Researcher, 08950 Barcelona, Spain

**Keywords:** ncRNAs, nanotechnology, oligonucleotide therapy, delivery vehicles, nanoparticles

## Abstract

Non-coding RNAs (ncRNAs) are emerging therapeutic tools but there are barriers to their translation to clinical practice. Key issues concern the specificity of the targets, the delivery of the molecules, and their stability, while avoiding “on-target” and “off-target” side effects. In this “ncRNA in therapeutics” issue, we collect several studies of the differential expression of ncRNAs in cardiovascular diseases, bone metabolism-related disorders, neurology, and oncology, and their potential to be used as biomarkers or therapeutic targets. Moreover, we review recent advances in the use of antisense ncRNAs in targeted therapies with a particular emphasis on their basic biological mechanisms, their translational potential, and future trends.

## 1. Introduction

Non-coding RNAs (ncRNAs) play an important role in the control of gene expression. Their ability to target multiple mRNAs at different levels within the same pathways or across different pathways suggests that they have great potential in the treatment of complex diseases. However, barriers to the translation of ncRNA-based therapies to the clinic are related to: (i) the specificity of the targets involved in the main pathways and contributing to disease development or progression, (ii) the local delivery of the molecules at the subcellular level, and (iii) their stability. In this special issue devoted to the potential use of “ncRNA in therapeutics”, we have compiled a series of articles about ncRNAs differentially expressed in cardiovascular diseases [1], in bone metabolism-related disorders [2], in epilepsy [3], and in oncology [4], that contribute to disease development or progression and have the potential to be used as biomarkers or targets for therapeutic development. Several clinical trials using various strategies have confirmed the success of ncRNA-based therapies [5]. These are, in broad terms, antisense therapies with small effectors (single- or double-stranded RNAs or DNAs) that follow classic Watson–Crick pairing rules for targeting different RNAs (mRNAS, miRNAs, lncRNAs). Recent research has led to the development of: (i) improved algorithms for a more effective and specific hybridization to target sequences, (ii) new chemistries for stabilizing effectors, and (iii) more efficient vehicles for specific targeting [6]. In this review, we discuss the suitability and limitations of the use of antisense ncRNAs in therapeutics, including the nature of tissue barriers, and examine a variety of current approaches for enhancing their delivery. We present technical advances to increase nucleic acid stability “in vivo”, in body fluids, and with respect to cell permeability, improve their tissue-specific targeting and avoid their potential off-target effects.

## 2. Strategies for Fine-Tuning Therapeutic ncRNA Levels by Using Specific Inhibitors or Activators

Recent developments in nucleic acid sequencing have highlighted the complexity of the RNA world, composed not only of protein-coding mRNAs, but also a plethora of regulatory RNAs named non-coding RNAs. These are classified internationally, based on their size, into short ncRNAs for those smaller than 200 nucleotides that are not translated into proteins, and long noncoding RNAs (lncRNAs). Short ncRNAs include micro-RNAs (miRNAs), small nuclear RNAs (snRNAs), small nucleolar RNAs (snoRNAs), transfer RNAs (tRNAs) and piwi-interacting RNAs (piRNAs). The lncRNAs comprise a highly heterogeneous group that are often involved in transcriptional repression of the proximal protein-coding gene and include intergenic transcripts (lincRNAs), enhancer RNAs (eRNAs), sense or antisense transcripts that overlap other genes, circular RNAs (circRNAs), or circular intronic lncRNAs (ciRNAs) [6]. Their involvement in several diseases has been reviewed in the special issue of “ncRNAs in Therapeutics”. Selene De Benedittis et al. [3] proposed three circulating miRNAs (miR-142, miR-146a and miR-223) as biomarkers that predict the clinical response to antiepileptic drugs. An “in-silico” approach using the “SpidermiR” R package generated a complex gene network identifying several genes with a central role in the drug resistance process. The genes highlighted were related to cell cycle control and apoptosis (CDK2 and PARP1), chromatin remodeling (SMARCD1, CARM1, PTBP2), transcription (STAT3, E2F1, SP1, and HIF1A), autophagy (SIRT1), inflammation (SOCS1, CXCL2, CCL3, IL6 NLRP3), and membrane transport (claudin1, CYB5A, CFTR, MDR1/ABCB1). The difficulty in obtaining brain tissue from drug resistant epileptic subjects makes it problematic to validate reported data. Michal Kowara et al. [1] reviewed miRNAs and lncRNAs associated with atherogenesis as potential targets to improve myocardial infarction prognosis. These authors summarized certain preclinical studies targeting miRNAs as regulators of crucial pathways involved in atherosclerotic plaque progression (miR-33, miR-98, miR-145, and members of the same miRNA cluster miR-494/miR-495), and other miRNAs indirectly associated with atherosclerosis (miR-29, miR-135b, miR-181b, miR-210, miR-335, miR-520c, and let-7g). Potential ncRNAs associated with cellular death (miR-133, miR-19a/19b, miR-494, miR-124, miR-325-3p), apoptosis (miR-15, miR-199a, miR-210, miR-34, miR-24, miR-199a, miR-210, miR-34), autophagy (miR-99a, miR-22, miR-144) during myocardial infarction, posterior tissue fibrosis (miR-21, miR-17a-3p, miR-590-3p), and angiogenesis (miR-34a, miR-26a, miR-378, miR-210) with consequences for cardiac remodeling, were also identified. For circRNAs, some details have been provided of circ-Ttc3, which exerts part of its effects by sponging miR-15b-5p, circFndc3b that reduces cardiomyocyte apoptosis, HRCRs (heart-related circRNAs) that sponge miR-233 and inhibit cardiac hypertrophy, and circ-FOXO-3 that promotes cellular senescence. 

These ncRNAs add additional layers of gene expression regulation and are promising targets for pharmacological intervention, via complementary base pairing, for treatment of diseases so far not susceptible to drug interventions. Therapies targeting short ncRNAs are based on the use of chemically synthesized nucleic acid polymers focused on gene silencing via inhibitors or gene activation with agonists. Most inhibitors used in the clinic, or currently in development, are RNA-based drugs, including synthetic single-strand nucleic acid polymers (antisense oligonucleotides or ASOs that are approximately 20 nucleotides in length) that inhibit mRNA translation and double-stranded RNA molecules (short interfering RNAs or siRNAs of 14 kDa), or that target mRNAs, miRNAs, or lncRNAs, inducing their degradation in the cytosol [7,8,9]. Furthermore, chemically modified double-stranded DNAs (dsDNAs) are popular as miRNA agonists (agomiRs), while steric-blocking ASOs (designed to bind target transcripts without becoming RNase H substrates) have been extensively used to competitively inhibit miRNAs (antagomiRs). Alternatively, therapeutic targeting of lncRNA expression can be achieved by other approaches. Targeting upregulated lncRNAs with ASOs or siRNAs can reverse their transcriptional gene repression effect. The expression of lncRNAs can also be modulated via steric blocking of the promoter or by using genome-editing techniques such as CRISPR/Cas9. Other strategies tested to assess lncRNA functions include the use of viral vectors, the generation of knockout mice with relevant corresponding natural antisense transcripts using CRISPR/Cas9 gene editing tools [10,11], and the generation of knock-in mice overexpressing lncRNAs. In a current review from this issue, Michal Kowara et al. [1] provide some examples of lncRNAs involved in atherosclerosis, such as MALAT1 (using double knock-out animals), lincRNA-p21 (through local injection of an siRNA-expressing lentiviral vector), or RAPIA (down-regulated by a short hairpin delivered through an adenoviral vector), whose activity is in part mediated by miR-183 inhibition. Other lncRNAs identified include those involved in apoptosis, such as KLF-3-AS1 (injection of mesenchymal stem-cell-derived exosomes) or MEG3, those related to the TGF-β pathway, such as a lncRNA called *Safe* and lnc-Ang362, those related to connective tissue growth factor (CTGF), such as lnc-n379519, and those involved in cardiomyocyte proliferation, such as CRRL (cardiomyocyte regeneration-related lncRNA) and CHRF (cardiac-hypertrophy-related factor). Cinzia Aurilia et al. [2] reviewed lncRNA–miRNA–mRNA crosstalk in bone metabolism-related diseases. These authors also reported a bioinformatic analysis that yielded 1017 genes and 662 lncRNAs differentially expressed in human mesenchymal stem cells during osteoblast differentiation, 496 genes and 24 lncRNAs in peripheral blood monocytes, when high versus low bone mineral density individuals were compared, and 10 mRNAs and 10 lncRNAs during osteoclast differentiation of CD14+ monocytes. In addition, some lncRNAs associated with primary bone tumors, such as GAS5 (growth arrest-specific transcript 5), TUG1 (taurine up-regulated gene1), and MALAT1 (metastasis-associated lung adenocarcinoma transcript 1), were reviewed. The authors pointed out that one of the major challenges for the potential application of ncRNAs in clinical practice is the few sequence similarities among species so that findings observed in animal models may not be directly transferable to humans. Finally, Riccardo Di Fiore et al. [4] reviewing miRNAs deregulated in rare gynecological cancers, proposed several over-expressed oncomiRs (miR-9, miR-10a, miR-21, miR-590-5p, miR-3147 and miR-4712), and some downregulated tumor suppressors (miR-34, miR-126, miR-196b), as potential targets for restoring or silencing key functions with therapeutic potential. These authors included information about GYNOCARE (European Network for Gynecological Rare Cancer Research) for the dissemination of recent medical and technological advances for clinicians and patients, with the aim of enabling access to possible participation in clinical trials. 

Regarding mechanisms, single-stranded molecules (ssRNAs or ssDNAs) cleave targeted sequences of complementary RNA via RNase H1-mediated degradation (recognizing RNA-DNA heteroduplex substrates), through steric blockade of translation or by modulating splicing [12]. Nucleic acids can also interact with proteins through the formation of 3D secondary structures (e.g., aptamers, single-stranded nucleic acid molecules that act as ligands that interact with target proteins through their 3D structure), or guide RNA molecules containing hairpin structures may bind to exogenously introduced Cas9 protein and direct it to specific DNA loci for targeted gene editing [13]. Since RNA is chemically unstable, its structural components need to be chemically modified to increase resistance to endonucleases. For example, antagomiRs and agomiRs include a phosphorothioate (PS)-backbone and 2′-O-Me-modified nucleotides complementary to a miRNA sequence [14,15]. Nevertheless, ASOs can be trapped in late endosomes, multivesicular bodies and lysosomes, reducing the amount of free, reactive molecules [16].

## 3. Physiological Barriers Hindering the Clinical Use of Oligonucleotide Therapies

A number of limitations have to be overcome to successfully translate ncRNA-based therapeutics into the clinic. These include: the induction of the inflammatory response caused by the recognition of exogenous RNAs by pathogen-associated molecular pattern (PAMP) receptors of the innate immune system; the difficulty in achieving efficient delivery for specific cell-targeting, avoiding undesired “on-target” effects due to uptake in cells other than the cells of interest; “off-target” interactions caused by either sequence similarities or overdosing to levels much higher than expected endogenously; sequence and chemistry-dependent toxicity; and saturation of endogenous RNA processing pathways.

Oligonucleotides display suboptimal pharmacokinetic and pharmacodynamic properties owing to their high molecular size (ASOs are ∼4–10 kDa and siRNAs are ∼14 kDa) and to their negatively charged phosphodiester backbone [17]. Their blood half-life is short (less than 5 min) as unmodified ASOs are easily degraded by endogenous nucleases in the extracellular space, particularly by 2′hydroxyl-dependent-RNases [18]. Another mechanism interfering with the activity of nucleic acid-based drugs is the presence of membrane barriers (Figure 1), i.e., the endothelial and the blood brain barriers, the renal clearance system and the reticuloendothelial system (mononuclear phagocytes, liver sinusoidal endothelial cells, and Kupffer cells (KC)). The relevance of the tissue barriers depends on the chemical and physical properties of the nucleic acid-based therapeutics employed. Molecules circulating in the blood can be transported across the endothelial barrier by two routes that are tightly regulated by various signaling systems: the paracellular transport that occurs through cell-junctions and is limited to molecules of less than 6 nm of diameter, or the caveolar-mediated transcytosis that carries large molecules within vesicles of 70 nm [19]. An alternative to pass the endothelial barrier is to administer therapy subcutaneously [20]. In addition, the endomembrane trafficking machinery plays an important role; the concept of an “endosome escape barrier” is one of the most important challenges for the effective use of nucleic acids in therapeutics [21]. Cationic lipids and polymers have been used to destabilize the endolysosomal barrier or to change intra-endosomal pH to alter endosome stability and trafficking. It is important to note that once anionic single-stranded oligonucleotides reach the cytosol they readily enter the nucleus [5]. To preserve their activity, systemically injected nucleic-acid-based drugs must also bypass renal clearance. The renal barrier filters molecules over 3-6 nm of diameter. The phosphorothioate (PS) backbone of chemically modified nucleic acids can bind to plasma proteins, thus increasing their size, allowing a broader tissue distribution, reducing renal clearance, and increasing the circulation time. Renal clearance is also correlated with the negative charge of the nucleic acid backbone [22], and novel approaches to oligonucleotide synthesis are being developed to mask the negative charge of the phosphate backbone of nucleic acids [23]. To date, most nucleic acid-based drugs have focused on either local delivery (e.g., into the cerebrospinal fluid via lumbar puncture) or delivery to the liver. However, the development of effective technology for extrahepatic systemic delivery remains a major goal.

A popular means of increasing nucleic acid stability is by binding to nanoparticles (NPs). However, the composition of their protein corona (formed in vivo by the adsorption of serum proteins) may constitute real ligands for cell surface receptors or complement activators, and they can be internalized preferentially by the mononuclear phagocytes of the reticuloendothelial system, which express high affinity Fc receptors and complement receptors [24]. The formation of a pronounced protein corona may be attenuated by PEGylation that reduces unwanted binding to liver KC and liver sinusoidal endothelial cells (LSEC), and by conjugation with CD47, which serves as “do not-eat-me” signal for macrophages [25]. 

Finally, exogenous nucleic acids can trigger inflammatory responses via interactions with pattern recognition receptors, including membrane-bound toll-like receptors (TLRs) that can induce the interferon response, or with cytosolic RIG-I family receptors [26]. Specifically, TLR3 recognizes ds-RNA motifs, TLR7 and TLR8 recognize ss-RNA, and TLR9 recognizes unmethylated CpG dinucleotides [13]. The most common strategy to skip the tissue barriers includes chemical modifications to the nucleic acid backbone to increase resistance to endonucleases, promoting binding to proteins that results in prolonged tissue retention and avoiding an immunological response.

## 4. Improving the Performance of Nucleic Acid-Based Therapies

### 4.1. Nucleic Acid Stability Enhancement

Several approaches have been proposed to improve the pharmacokinetics and stability of oligonucleotide drug delivery, and to reduce their immunogenicity and cytotoxicity. These include new chemical modification of the nucleic acid backbone (the ribose sugar moiety and the nucleobase), or using improved NPs binding ASOs to receptor targeting agents (Table 1 and Figure 2) [13]. Furthermore, resistance to endonucleases can be increased by modifying the phosphorothioate (PS) bonds in therapeutic oligonucleotides, by replacing oxygen atoms in the ribose backbone by a sulfur group or modifying the 2′ position of the ribose sugar (2′-O-methyl, 2′-O-methoxyethyl, and 2′-fluoro, are the most common substituents). Using bridge nucleic acids, such as locked nucleic acids (LNAs) in which the 2′-O and the 4′-C on the same ribose are linked by a methylene bridge (antagomiRs), improves nuclease stability and the affinity of the oligonucleotide for its target RNA [27]. Interestingly, it has been reported that 2′-ribose modifications to inactivate the passenger strand of the siRNA duplex can reduce both immunostimulatory and off-target effects [5]. A disadvantage of PS modifications is the reduction of the binding affinity of the oligonucleotide for its target. In addition, siRNAs that contain PS modifications at every linkage are less active than their equivalent phosphodiester-based siRNAs [13].

The efficacy of nucleic acid delivery can be enhanced through direct covalent conjugation of various moieties to lipids such as cholesterol (endocytosis of cholesterol-conjugated siRNAs are mediated by SR-B1 or LDL-R), cell-penetrating peptides (CPPs), aptamers (considered chemical antibodies, because of their binding to target proteins with high affinity), antibodies (specific interactions with a cell surface receptor facilitates their delivery to cell subpopulations that are not accessible using other technologies), and sugars (e.g., N-acetylgalactosamine, which binds to asialoglycoprotein receptor-1, highly expressed in the liver) [13].

### 4.2. Nanoparticles as Delivery Vehicles: Advantages and Limitations

The development of a universal drug delivery system (DDS) based on nanoparticles (NPs) is a primary research field in nanotechnology, especially for efficient encapsulation and controlled release (Figure 3). The main advantages of nanoparticles are the tailored optimization of biophysical and biological properties, allowing the creation of highly customized delivery platforms. The use of NPs has risen exponentially due to their wide range of biomedical applications; however, their potential adverse consequences for human health must be also considered. Biocompatible NP carriers offer protection against extracellular degradation by nucleases, either by dense compaction or encapsulation of nucleic acids [28]. Important considerations in the design of NPs are their size, nature, and surface characteristics, as not all of them are suitable for clinical applications. Biocompatibility or biodegradability is required to release their cargo at the target site and to reduce or avoid potential risks. Biodistribution studies have detected the accumulation of NPs of large size (more than 200 nm diameter) in lungs, and in the liver, when they are administered systemically, while NPs of less than 8 nm are cleared by the kidneys [29]. The clearance function of the liver is conferred by KC and LSEC, which are equipped with several receptors, including different C-type lectin receptors, such as the mannose receptor CD206, and scavenger receptors that generally bind negatively charged ligands. Biliary clearance is observed especially for particles over 200 nm, and for strongly charged particles [30]. After subcutaneous administration, small NPs are easily transported into lymph nodes, whereas larger particles remain at the site of administration [31]. Lastly, dendritic cells efficiently internalize particles larger than 200 nm [32], whereas monocytes and macrophages internalize larger particles by receptor-mediated endocytosis and phagocytosis [33].

The shape of the NPs may also affect the efficacy of uptake. Thus, spherical NPs are internalized more efficiently by macrophages than elongated NPs [34]. Coating with cell- penetrating peptides (CPP) can increase the internalization of NPs [35].

### 4.3. Nanoparticles as Delivery Vehicles: Effects of Structure and Composition

A large variety of materials and structures have been evaluated for the safe transfer of nucleic acids (Table 2 and Appendix A), including different inorganic materials, non-covalent complexation with cationic polymers, dendrimers, and CPPs. We next focus on lipid-based formulations (lipoplexes and liposomes) since they represent one of the most common approaches for delivery. 

Nucleic acids encapsulated into lipid-based nanoparticles (LNPs), built on a mixture of four components (cationic or ionizable lipids for RNA complexation, neutral lipids to stabilize NPs, helper phospholipids to aid formation and intracellular release, and PEGylated lipids or PEI-based lipids to reduce non-specific interactions), are the most tested delivery vehicles in both pre-clinical and clinical studies [52,53].

Neutral helper lipids, such as cholesterol, result in much stronger transfection efficiency likely due to elevated endosomal escape of passenger nucleic acids, and the incorporation of coiled-coil lipopeptides into liposomes, resulting in the direct release of charged nucleic acids into the cytosol. Advantages of these neutral helper lipids include: (i) a more specific and efficient cellular internalization, since complexing negatively charged nucleic acids with cationic lipids promotes interaction with the negatively charged cell membrane [63], (ii) the triggering of endosomal escape, and (iii) increasing stability and protection from degradation in extracellular spaces, with the functional secondary folding and 3D structure preserving the steric accessibility of the functional RNA domains. However, the use of cationic lipids is also associated with toxicity since they can disrupt the integrity of cell membranes and induce vacuolization of the cytoplasm [64]. In addition, cationic lipids can interact with negatively charged serum proteins to form aggregates which are eliminated by the liver and the spleen. Several strategies to reduce the cationic charge have been attempted, such as the use of cholesteryl oleate in cationic solid- lipid-nanoparticle(cSLN)-nucleic acid formulations to improve cytotoxicity. In addition, ionizable lipids (a class of lipids bearing neutral or mild positive charge at physiological pH exposing high cationic groups in acidic conditions within endosomes, facilitating endosomal escape) are preferred to cationic lipids [65]. LNPs can be further functionalized with ligands that confer cell-specific targeting. However, an increase in complexity complicates manufacture and may increase their toxicity, which is a major concern that may limit their clinical utility.

## 5. Future Trends in the Clinical Use of Nucleic Acids for ncRNA Therapy

The successful application of RNA-based therapies requires an interdisciplinary approach to improve tolerability, specificity and delivery, and several strategies are currently under development.

### 5.1. Avoiding Immune-Related Adverse Reactions: Nucleic Acid–TLR Interactions 

Not all miRNAs induce similar immunogenicity, and the difficulty in prediction of this response has prompted the use of screening methods in preclinical studies. Since immune responses differ between animal models and humans, such screening methods should employ human cells. Primary cells are preferred as cell lines may have impaired response pathways. The use of co-culture and organoid systems [66], as well as patient-derived xenograft models [67], 3D cell culture models [68], or “organs-on-a-chip” [69], could provide a better assessment of systemic response. In addition, a database of miRNAs targeting TLRs has been proposed, specifying the exact immune adverse reactions and the severity of symptoms, allowing the selection of therapeutic RNAs with the smallest potential immunogenicity before the initiation of clinical trials [70].

Lastly, since an efficient activation of TLRs requires a length of at least 21 nucleotides of ssRNA, the use of smaller RNAs such as LNA antimiRs, with a short sequence of 7-8 nt, termed “tiny” LNAs, has been proposed [70]. These LNAs target the 5′-seed region of miRNAs and can inhibit an entire miRNA family sharing such a seed region, increasing the potential for off-target effects. Nevertheless, and despite success in preclinical studies, “tiny” LNAs have not yet been clinically assessed. 

### 5.2. Improving the Specificity of Targeting Nucleic Acids

Targeted therapy is challenging. Several approaches have been suggested to avoid “on-target” side effects, due to non-specific cell uptake, and “off-target” side effects, such as expressing therapeutic RNAs under a specific promoter to restrict expression to the cells of interest. Nevertheless, in such cases dosing should be carefully monitored since several studies indicate that shRNA overexpression by strong promoters could cause neurotoxicity owing to saturation of the RNAi machinery [71].

A strategy to prevent the harmful side effects caused by the delivery vehicle is to link the targeting RNA to a ligand whose receptor is overexpressed in the cells of interest. The key beneficial effect relates primarily to increased uptake at the cellular level rather than overall changes in biodistribution [72]. This strategy is particularly suited to target receptors overexpressed in cancerous cells since they deliver the cargo directly into tumor cells [73], e.g., by conjugating ASOs to N-acetylgalactosamine (GalNAc), which binds to the high capacity asialoglycoprotein receptors (ASGPR) in the liver [74].

Another strategy, reported by Michal Kowara et al. in this collection [1], employs a multifunctional biomimetic nanoparticle system, ternary polyplexes coated with ApoA-I resembling HDL particles, which interacts with specific receptors on macrophages and creates a positive feedback loop facilitating drug delivery to the macrophage. Polyplexes employ multi-platforms targeted at different cells such as core-shell NPs, composed of a PGLA core and three external layers, a lipid layer, an ApoA-I layer for enhanced entry into macrophages, and a hyaluronic acid layer for endothelial cell targeting. However, currently there are alternative non-viral systems for in vivo targeting of other cell types with high cardiovascular interest such as cardiomyocytes or cardiac fibroblasts [75].

### 5.3. Improving the Delivery of Nucleic Acid-Nanoparticle Systems

Once specificity is achieved, efficient intracellular delivery of the cargo, e.g., the ability to exit from the endomembrane system, is one of the greatest challenges in the field, with the foremost reason for premature clinical trial termination being the lack of efficacy of current delivery methods [13]. The endomembrane trafficking machinery plays a key role in the potential use of ncRNA-based therapies and increasing the understanding of its mechanistic basis will provide important clues to improve their clinical success [21]. Numerous strategies have been devised to facilitate endosomal escape, with exosome-mediated delivery of nucleic acids (mostly miRNAs) being a promising technical development.

Exosomes are heterogeneous extracellular cell-derived phospholipid nanovesicles of ∼100 nm in diameter that can deliver bioactive molecules to specific recipient cells [76]. Exosomes are natural carriers of miRNAs and may present an ideal delivery system owing to their negligible antigenicity and lack of toxicity. The precise uptake mechanism for exosomes is not fully characterized, but it seems to circumvent phagocytosis and by-pass the endocytic pathway [13]. Bioengineered exosomes provided with improved target-homing specificity, stability, and capability of overcoming in vivo barriers, can lead to a more specific and efficient delivery of molecules and can serve as a modular platform on which combinations of therapies and/or targeting motifs can be loaded [77]. However, prior to ensuring their safe clinical use, much research is needed to reduce the immunogenicity of allogeneic exosomes enriched in major histocompatibility complex proteins with the associated risk of causing immune reactions. Other important challenges to the clinical use of exosomes are the difficulties associated with their large-scale production and high manufacturing costs [78].

Much effort is also being dedicated to the development of NPs able to co-deliver different therapeutic agents to target multiple genes in multiple cells as combinatorial therapeutics. Polymeric micelles [79] and circulating NPs with a pH-regulated drug release mechanism have shown promising results in mouse models [80]. Another promising approach is the development of bio-engineered RNA molecules capable of delivering multiple small RNAs at once, including siRNAs, antimiRs, and miRNA mimics. These constructs have successfully inhibited the growth of multiple lung cancer cell lines in vitro [81].

Alternatively, novel approaches, such as the creation of advanced biomimetic materials involving bacterial and viral-based NPs (outer membrane vesicles and virus-like particles) [82], or intelligent DNA nanomachines, either naked or modified with ligands [83], show promise for targeted or personalized treatments. 

Finally, smart strategies involving stable peptide-coated nanocarriers for oral delivery are also emerging, although their delivery remains a challenge [84]. Nevertheless, disruptive technologies such as the ingestible robotic biologic pill (RaniPill™ capsule) will likely enhance the oral bioavailability of all kinds of biologic drugs including ncRNAs. 

### 5.4. Selecting Appropriate Patients

Stringent patient selection remains a key factor for successful translational science. Thus, a precise definition of trial patient cohorts based on a single molecular mechanism is required [75]. In addition, these stratifying strategies suggest that therapy for many rare or currently untreatable diseases will be possible through use of precision genetic medicine [13].

### 5.5. Integrating Complex Regulatory Systems

The human genome/epigenome could be considered as a concatenation of regulatory elements that work together in an integrated way. Thus, any modification in one of these regulatory nodes will have an impact on the rest of the regulome. This implies the need for more understanding of the interactions among these regulatory nodes, including the identification of yet unrecognized “master epigenetic regulators” that might prove useful as therapeutic targets [75]. The application of artificial intelligence and computer modelling can help to identify the interactome of the disease in a specific patient and to select the best delivery technology to guide a personalized nucleic acid-based treatment.

## 6. Conclusions

Barriers to the translation of nucleic acid-based therapeutics into the clinic are related to stability, specificity, delivery, and toxicity issues. Thus, there is a need for targeted therapies to achieve precision medicine, avoiding “on-target” and “off-target” side effects. The main approaches for therapeutic targeting based on nucleic acids include the use of proteins, gene editing, and cell therapy through NPs. Several investigations regarding potential targets are being undertaken in animal models to test the effective delivery of oligonucleotides to their intracellular sites of action. Lack of efficient delivery of ncRNAs remains one of the greatest challenges to address and the foremost reason for premature clinical trial termination. Nevertheless, because of their versatility, nucleic acid-based technologies will soon become an integral part of the therapeutic armamentarium for the treatment of both common and rare diseases. 

## Figures and Tables

**Figure 1 ijms-22-11596-f001:**
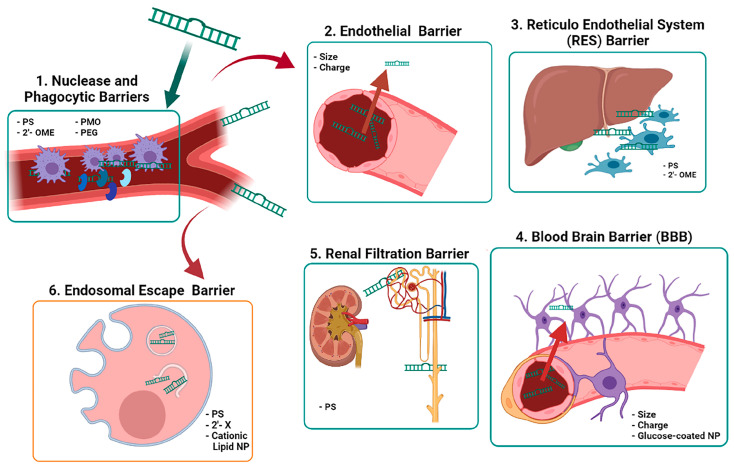
Extracellular and intracellular barriers for therapeutic ncRNAs action. The diagram shows systemic delivery and metabolism of therapeutic ncRNAs (**1**–**5**), as well as intracellular uptake (**6**). Chemical modifications and improvements to avoid nuclease degradation are listed inside the frame as hyphens and small letters. Abbreviations: 2′-X: general modification at 2′; 2′-OME: 2′-O-methyl; PS: phosphorotioate; PMO: phosphorodiamidate morpholino oligomers; PEG: pegylation; NP: nanoparticles. Image created with BioRender.com.

**Figure 2 ijms-22-11596-f002:**
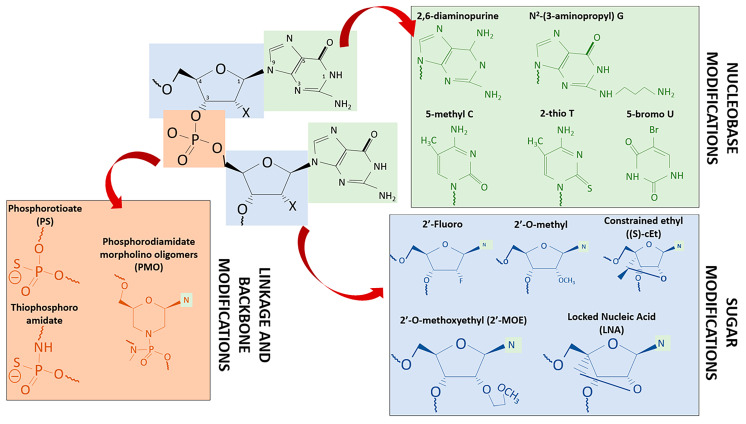
ASO chemical modifications. Key chemical modifications of ASOs are site-specific, based on the structure, as shown in this example of an RNA molecule with dinucleotide purines, linkage, and sugar numbering, where X could be H (for DNA) or OH (for RNA). Relevant modifications from each group used in clinical trials and commercialized drugs are shown with their corresponding color (green for nucleobase modifications, blue for sugar modifications, orange for phosphodiester linkage modifications, “N” with green halos correspond to simplified nucleobases). Modification improvements are listed on Table 1.

**Figure 3 ijms-22-11596-f003:**
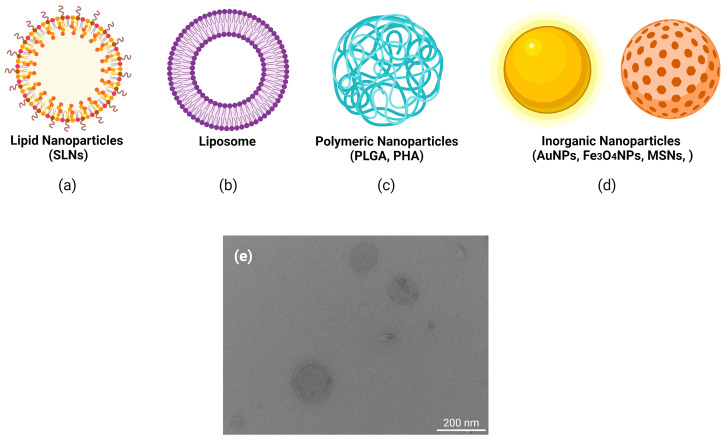
Main types of nanoparticles used in clinical trials for the delivery of nucleic acids. Different type of nanoparticle formulations designed for tissue targeting in order to achieve therapeutical effects. (**a**) lipid nanoparticles (e.g., solid lipid nanoparticles, SLNs), (**b**) liposomes, (**c**) polymeric nanoparticles (e.g., polylactide-coglycolide, PLGA; polyhydroxyalkanoates, PHAs) (**d**) inorganic nanoparticles (e.g., gold nanoparticles, AuNPs; iron nanoparticles, Fe_3_O_4_NPs; mesoporous silica nanoparticles, MSNs), (**e**) transmission electron microscopy (TEM) image of cholesteryl oleate SLNs, fabricated at the Faculty of Pharmacy and Food Science (Universitat de Barcelona), showing the scale bar in nm. Image created with BioRender.com.

**Table 1 ijms-22-11596-t001:** Antisense oligonucleotide modifications and improved activities for clinical purposes.

Chemical Modification	ASOs Improvement	Commercialized or Phase 3 ASOs Drug
**Nucleobase Modifications**		
2,6-diaminopurine	Enhance electrostatic interactions with phosphate backbone Enhance target binding affinity and specificity Enhance duplex thermal stability	
N2-(3-aminopropyl) G		
5-methyl C		
2-thio T		
5-bromo U		
**Sugar Modifications**		
2′-Fluoro	Shows duplex stabilizing properties and binding to dsDNA Enhance binding affinity for target RNA sequences Reduce susceptibility toward nuclease degradation	
2′-MO		
(S)-cEt		
LNA		Miravirsen
2′-MOE		Mipomersen, Nusinersen, Volanesorsen
2′-H		Fomivirsen, Mongersen
**Phosphodiester Linkage & Backbone Modifications**		
Phosphorotioate (PS)	Improvement of resistance to nuclease cleavage Enhance binding to albumin and heparin proteins Improvement in cellular uptake	
Thiophosphoroamidate		
Phosphorodiamidate morpholino oligomers (PMO)		Eteplirsen, Golodirsen

Key chemical modifications of each structure are shown. Stability against nucleases, binding affinity, and specificity are the main improvements made in ASOs allowing them to enter clinical trials. Although hundreds of ASOs are currently under clinical trials, only those that are commercialized or clinically advanced in phase 3 are shown here. Many of them combine several modifications; therefore, clinical ASOs are listed including only the most relevant modifications. Abbreviations: 2′-H: 2′-deoxy; 2′-MO: 2′-O-methyl; 2′-MOE: 2′-O-methoxyethyl; (S)-cEt: constrained ethyl; LNA: locked nucleic acid; PS: phosphorotioate; PMO: phosphorodiamidate morpholino oligomers; G: guanine; C: cytosine; T: thymine; U: uracil.

**Table 2 ijms-22-11596-t002:** Materials and structures evaluated for the transfer of nucleic acids.

SMaterials	Properties	Current 2021 Clinical Trials	Toxical Profiling	References
**Inorganic**				
Noble metal (Au, Ag, Pt) NPs	Biocompatible, surfaces with multiple cargo,	SP1–SP4	Cytotoxicity, inflammation, apoptosis	[36,37,38]
Silica	inmunotherapy	SP5	Cytotoxicity dose dependent. Oxidative stress, inflammation	[39,40]
Iron oxide (IONPs and SPIONs), Ferritine	Biocomptability, wide range of sizes and shapes	SP6–SP10	Cytotoxicity	[41,42]
**Carbon nontubes**				
Graphene base nanomaterials	Large surface area, high charge carrier mobility and high stability	SP11–SP14	Cytotoxicity dose dependent, particle aggregation	[43,44]
**Organic polymers**				
** *Proteins-stabilized NPs* **				
Albumin	Biocomptability, facilitate endocytosis, great loading efficiency	SP15–SP51	Low	[45]
Collagen	Biocompatible, control drug releasing	Non	Low	[46,47]
Gelatin	Biocompatible, biodegradable	Non	Low	[48]
CPPs (Cell Penetrating peptides)	Translocate across biological membranes	Non	Low	[49]
** *Polysaccharides* **				
Chitosan	Biocompatible, biodegradable, sustain drug release, low immunogeneity	SP52	Low	[50]
Alginate	Biocompatible, low immunogeneity	Non	Low	[51]
** *Lipid-based nanoparticles (LNPs)* **	Enhance internalization and endosomal scape	SP53–SP62	Disruption of cell membranes and protein aggregation	[52,53,54]
**Covalent complexation with polymers**				
PLGA (poly-D,L-lactic-co-glycolic acid)	Biocompatible, biodegradable.	Non	Low	[55]
PEG (polyethylen glycol)	Increase circulation time and efficiency	Non	Immune-mediated side effects	[56]
PEI (polyethylenimine)	“Proton sponge” and facilitate endosomal scape.	Non	Oxidative stress and DNA damage.	[57]
Poly-L-glutamate	Biocompatible	Non	Low	[58,59]
Dendrimers	Well physical characterized	Non	Oxidative stress and DNA damage.	[60,61]
Charge-altering releasable transporters (CARTs)	Endosomal scape	Non	No tested	[62]

Further information about current clinical trials (SP1–SP62) listed in Table 2 are summarized in Appendix A. For more information on references [60,61,62,63,64,65,66,67,68,69,70,71,72,73,74,75,76,77,78,79,80,81,82,83,84], see the references section.

## Data Availability

Not applicable.

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
