# Peer review of "ncRNAs in Therapeutics: Challenges and Limitations in Nucleic Acid-Based Drug Delivery"

_ijms, 2021, doi:10.3390/ijms222111596_

Round 1

Reviewer 1 Report

Manuscript by Hueso et al. is a neat introduction to special issue concerning ncRNAs in therapeutics. I have several minor comments which may help to improve the clarity of the message for the readers.

- I suggest to rewrite a bit an Abstract or first part of Introduction, as information and sentences there are too similar.

- I find some information given not clear, e.g. „specificity of the targets through the main pathways” (Abstract), „on-target side effects” (Abstract, Introduction and 3.chapter). I recommend including such Information once and there where is the place to explain this in more details.

- Abbreviations for various types of short ncRNAs should be explained when given for the first time. (2. Chapter).

- circRNAs are not to be classified as short RNAs (2. Chapter).

- Information about siRNAs and ASOs molecular mass in kDa is given twice.

- Information about ASO and siRNAs mechanism of action is confusing (2. Chapter, “The majority of inhibitors used…”). Degradation in cytosol is specific for siRNAs and as in the first part of the sentence ASOs are included, it would be worth for clarity to mention degradation initiated in nucleus by ASOs. Separation of information for siRNAs and ASO could also be considered.

- Information given about restoring lncRNAs function is not clear (2. Chapter, “Other strategies to restore lncRNAs function…”). I find strategy to restore lncRNA function as therapeutic one, while generation of knockout mice is a strategy to reveal lncRNA function.

- Please correct writing for RNase (not RNAse).

- Figure 2, maybe it would be worth to include blood-brain barrier here?

- Table 2 should be reformatted to be more readable.

Author Response

I appreciate the point of view of the reviewer and the suggestions for improving the paper.

Reviewer 1

Manuscript by Hueso et al. is a neat introduction to special issue concerning ncRNAs in therapeutics. I have several minor comments which may help to improve the clarity of the message for the readers.

1- I suggest to rewrite a bit an Abstract or first part of Introduction, as information and sentences there are too similar.

R: I have rewritten both sections to avoid repetitions.

2- I find some information given not clear, e.g. „specificity of the targets through the main pathways” (Abstract), „on-target side effects” (Abstract, Introduction and 3.chapter). I recommend including such Information once and there where is the place to explain this in more details.

R: Repetitions have been deleted and information has been expanded.

3- Abbreviations for various types of short ncRNAs should be explained when given for the first time. (2. Chapter).

R: Abbreviations have been explained the first time they were used.

4- circRNAs are not to be classified as short RNAs (2. Chapter).

R: Thank you for the warning. The classification has been revised.

5- Information about siRNAs and ASOs molecular mass in kDa is given twice.

R: Repeated information has been removed.

6- Information about ASO and siRNAs mechanism of action is confusing (2. Chapter, “The majority of inhibitors used…”). Degradation in cytosol is specific for siRNAs and as in the first part of the sentence ASOs are included, it would be worth for clarity to mention degradation initiated in nucleus by ASOs. Separation of information for siRNAs and ASO could also be considered.

R:This information has been clarified. Now the text says “The majority of inhibitors used in the clinic or currently in development are RNA-based drugs, including synthetic single-strand nucleic acid polymers (antisense oligonucleotides or ASOs that are approximately 20 nucleotides in length) that inhibit mRNA translation and double-stranded RNA molecules (short interfering RNAs or siRNAs of 14 KDa) that target mRNAs, miRNAs or lncRNAs inducing their degradation in the cytosol.

7- Information given about restoring lncRNAs function is not clear (2. Chapter, “Other strategies to restore lncRNAs function…”). I find strategy to restore lncRNA function as therapeutic one, while generation of knockout mice is a strategy to reveal lncRNA function.

R: I agree with the reviewer's comment and this point has been clarified. The text has been rewritten as ” On the other hand, therapeutic targeting of lncRNA expression can be reached by different approaches. Targeting upregulated lncRNAs with ASOs or siRNAs can reverse their transcriptional gene repression effect. LncRNA expression can be also modulated via steric blocking of the promoter or by using genome-editing techniques such as CRISPR/Cas9. In contrast, other strategies tested to reveal lncRNAs function include the use of viral vectors, the generation of knockout mice of relevant corresponding natural antisense transcripts using CRISPR/Cas9 gene editing tools [10, 11] or the generation of knock-in mice overexpressing the lncRNA”.

8- Please correct writing for RNase (not RNAse).

R: The word has been corrected

9- Figure 2, maybe it would be worth to include blood-brain barrier here?

R: I agree with the proposal and the BBB has been included.

10- Table 2 should be reformatted to be more readable.

R: The column in the medical application has been removed to make the graphics more user-friendly.

Reviewer 2 Report

The manuscript presents introductory notes for the special issue on the applications of noncoding RNAs as potential therapeutic agents. It provides a concise review of the progress in the area focusing on modern therapeutic strategies, potential difficulties in their applications, strategies of delivery and improving the specificity and efficiency of RNA-based therapeutics. In the last part the authors identify areas for future development. The paper is very well written in clear language, providing up to date overview of all the aspects of the ncRNA-based therapies. 

Author Response

 Reviewer 2

The manuscript presents introductory notes for the special issue on the applications of noncoding RNAs as potential therapeutic agents. It provides a concise review of the progress in the area focusing on modern therapeutic strategies, potential difficulties in their applications, strategies of delivery and improving the specificity and efficiency of RNA-based therapeutics. In the last part the authors identify areas for future development. The paper is very well written in clear language, providing up to date overview of all the aspects of the ncRNA-based therapies. 

R: Thank you for your comments.